# Real-World Data Difficulty Estimation with the Use of Entropy

**DOI:** 10.3390/e23121621

**Published:** 2021-12-01

**Authors:** Przemysław Juszczuk, Jan Kozak, Grzegorz Dziczkowski, Szymon Głowania, Tomasz Jach, Barbara Probierz

**Affiliations:** 1Systems Research Institute, Polish Academy of Sciences, Newelska 6, 01-447 Warsaw, Poland; 2Faculty of Informatics and Communication, Department of Machine Learning, University of Economics in Katowice, 1 Maja 50, 40-287 Katowice, Poland; jan.kozak@ue.katowice.pl (J.K.); grzegorz.dziczkowski@ue.katowice.pl (G.D.); szymon.glowania@ue.katowice.pl (S.G.); tomasz.jach@ue.katowice.pl (T.J.); barbara.probierz@ue.katowice.pl (B.P.)

**Keywords:** entropy measure, real-world data, preprocessing, decision table, classification

## Abstract

In the era of the Internet of Things and big data, we are faced with the management of a flood of information. The complexity and amount of data presented to the decision-maker are enormous, and existing methods often fail to derive nonredundant information quickly. Thus, the selection of the most satisfactory set of solutions is often a struggle. This article investigates the possibilities of using the entropy measure as an indicator of data difficulty. To do so, we focus on real-world data covering various fields related to markets (the real estate market and financial markets), sports data, fake news data, and more. The problem is twofold: First, since we deal with unprocessed, inconsistent data, it is necessary to perform additional preprocessing. Therefore, the second step of our research is using the entropy-based measure to capture the nonredundant, noncorrelated core information from the data. Research is conducted using well-known algorithms from the classification domain to investigate the quality of solutions derived based on initial preprocessing and the information indicated by the entropy measure. Eventually, the best 25% (in the sense of entropy measure) attributes are selected to perform the whole classification procedure once again, and the results are compared.

## 1. Introduction

In present times, we are facing the problem of a large amount of data flowing from different sources. In the era of the Internet of Things (IoT) and big data, the challenge is to effectively use and present the acquired data without generating redundant information. Due to the size of data available for decision-makers, it is nearly impossible to manually make any complex decisions. This difficulty is experienced even in machine learning algorithms, which must manage too many attributes, variables, and additional constraints, resulting in the whole process being lengthy and complicated [1]. As such, it is essential to simplify data in the cases where the decisions should be made very quickly, and a need exists to use a decision support system to maintain the decision-maker’s sovereignty.

The main drawback of the existing datasets is their uniform structure. For the data related to a single domain, the distribution of attribute values, the size of data, or the overall difficulty of the given dataset classification is expected to be on a similar level. However, in the case of more general approaches, we often face inconsistency in data, including the need to use additional knowledge from the domain experts. In general, data available in repositories are mostly preprocessed and directed on a particular problem (like the classification or the regression). At the same time, the initially collected data may still be very complex.

The above problem had led to the construction of many complex algorithms and methods intending to decrease the complexity of the data used in the decision process. Among these methods, we can emphasize approaches for reducing the number of variables included in the algorithm [2,3]. The idea of initially preprocessing the data related to the feature selection, removing the redundant data, or including more general attributes replacing the existing ones is not a new concept and it was deeply studied in many articles, where initial data limitation was needed. Examples of such feature selection methods can be found, for example, in extensions of the Principal Component Analysis method. One of the newest review articles in this subject can be found in [4]. A more general approach for future selection involving the swarm methods is presented in [5,6]. In comparison, one of the newest review articles related to the swarm methods is [7]. The second large set of algorithms used for the feature selection is related to the tree-based methods. In these methods, the attributes can be selected based on the importance of the attribute in the process of building the tree (classifier). An example of comparison for such algorithms can be found in [8].

For many cases, data dependencies are not linear. Thus, a complex method of variables elimination should be applied. For example, in the case of periodically important variables or in situations where the linear dependencies between elements are not obvious, different methods must be used to emphasize the crucial variables in the system. To avoid redundancy in the data, the selected variables should exhibit little or no mutual correlation. This requirement was described by [9], in which the phenomenon of the illusion of validity occurs: people have confidence in the results, which are based on redundant data. Thus, in decision support systems and during attribute selection, the role of decision-makers can also be marginalized.

A method that effectively identifies the crucial variables present in the complex data can be essential for the whole system’s efficiency. However, in the case where the data structure and its complexity makes the data difficult or even impossible to process, the decision-maker faces a two-step problem: First, there is a need to adapt the data to fit the algorithm’s input format. This can be achieved by some additional preprocessing methods, leading to a data format acceptable as the algorithm’s input. However, the whole process may be lengthy and complex. It often covers concepts such as filling the missing data, discretization, and scalarization. Dealing with missing data cannot be solved with simple methods, and the literature covers various approaches to this problem [10,11,12].

Thus, today we observe many algorithms dedicated to a particular domain, which, opposite to the general approaches, can deal with the problems more efficiently. However, one should know that such available methods can still be beneficial, even as a starting point for emerging domains related to complex or big data. Our idea was to collect raw data from different fields and prepare it in a uniform, easy-to-analyze format based on decision tables. At the same time, we tried to use as general tools as possible, which unfortunately can lead to a decrease in classification quality. However, it maintains the generalized approach for all datasets.

Furthermore, we selected entropy as a concept, which allows us to describe the disorder of the data. By the disorder, we understand here the measure of complexity, where the more complex data (fewer dependencies between objects and attributes is visible) is defined by the higher entropy values. Therefore, we assumed that the increase in entropy could be equated with data difficulty. Furthermore, this assumption is verified by performing the actual classification on various datasets. Eventually, the results from the classification on the full set of attributes and subset generated on the basis of entropy can be compared. It is expected that high entropy should lead to less effective classification.

The entropy measure is considered from the point of view of all attributes. Thus, it is possible to identify the attributes with small disorder values (smaller entropy values). A subset of attributes with small entropy could be used to perform the classification while the data is limited.

In our data, a clear distinction exists between conditional attributes and decision class. Data from various fields cover several objects as well as different numbers of attributes. However, the common goal is to perform a classification task on the presented data. The second step of our research completely focused estimating the impact of the entropy-based measure on the classification task. First, we tried to determine if entropy can be effectively used to indicate data difficulty. Eventually, we investigated the results of the classification of the data. We expected that, initially, all conditional attributes analyzed in the dataset could be treated uniformly (i.e., have similar entropy values). Thus, the main questions were: is there a correlation between the entropy values and the quality of classification, and can the entropy-based measure be used to select the best-fitted attributes for the classification problem? To summarize, our research steps were as follows:initial preprocessing of real-world data;entropy calculation for different datasets;classification on all datasets;selection of the best-fitted 25% of attributes based on the lowest entropy measure;the comparison between the classification results for the full and limited set of attributes on different datasets.

To generalize our observations as much as possible, we tried to select data from various fields and describe the whole preprocessing framework with the use of domain knowledge presented by experts from different fields. Moreover, this preprocessing schema allowed us to use a general data format, which can be effectively used in entropy calculation and, finally, in classification problems.

The paper is organized as follows: In the next section, we present the related studies. In Section 3, we discuss the theoretical background related to the subject, including a description of entropy, decision tables, and efficiency measures used in classification tasks. Section 4 contains a description of the real-world data covering different domains. Section 5 presents the results of our experiments based on entropy calculation as well as the classification problem. Eventually, we conclude the study in Section 6.

## 2. Related Works

By classical entropy, we understand the measure of uncertainty related with some data. The idea was introduced by Shannon in 1948 [13] and further extended, for example, by Renyi and Tsallis [14,15], where Renyi entropy is the generalization of the Shannon entropy for specific parameters.

The classical entropy measure is used as a crucial element in many different algorithms and methods. Amongst the most prominent examples are the well-known classification algorithm C4.5 developed by Quinlan [16] as an extension of algorithm ID3 [17]. In both examples, entropy was used as a measure to generate a classifier (a decision tree). In C4.5, entropy was used for all algorithm steps to calculate the information gain based on the entropy for every attribute available in the dataset. A similar idea is used in greedy heuristic ID3, where, once again, the attribute used as a split criterion for the data is based on the highest information gain. Such an approach has been successfully used in machine learning [18] and signal processing [19].

Entropy is often used as an element of broader methods rather than a standalone measure. It has a role in novel metaheuristics such as an extension of classical particle swarm optimization [20]. In [21], it was used as an alternative approach to the concept of fuzzy sets to measure the uncertainty of the task in a task assignment problem. Entropy was used as an extension of the binary classification problem solved by particle swarm optimization [22]. In many articles, entropy has often been used as a replacement for classical measures such as variance [23].

Entropy mixed with the concept of fuzzy sets was included in an outlier detection approach [24]. In [25], entropy was included as a part of the feature selection mechanism based on fuzzy sets. Finally, a more complex approach, including the fuzzy multicriteria approach based on the TOPSIS method, was presented in [26].

Entropy was used in many different approaches to measure randomness in a clinical trial [27]. In [28], entropy was introduced to measure the uncertainty of ordered sets. In general, it can be used as an idea of measure for different fields such finance [29,30], chemistry [31], physics [32], and more. However, no works used entropy as a general measure for different domains simultaneously. A separate direction of research is devoted to various extensions of classical entropy. In [33], the idea of measuring an entropy on different scales (multiscale entropy) was presented. In the case of time-series data, the concept of approximate entropy is often used [34]. In [35], approximate entropy was extended, called sample entropy. This idea was further extended in [36]. Both methods were used in different applications to address various dynamic aspects of systems.

Another prevalent extension of the classical measure is permutation entropy, effectively used as a nonlinear measure in different fields such as cyber-security [37] and fault diagnosis in systems [38]. Some preliminary comparisons between the classical entropy measure and Pearson correlation were introduced [39]. In this example, the authors focused on the data derived from the system from the Internet of Things, focusing on spatio-temporal data.

The idea of using entropy as a complexity measure is well-known, and it has been recently studied by many researchers. Among interesting examples, we mention [40], where information entropy was used to measure the genetic diversity in colonies. Another example covers the general idea of measuring the complexity of time series [41].

Entropy as a measure of diversity was presented in [42], where the authors used Shannon entropy to measure the urban growth dynamics for a case study related to real-world data from the city of Sheffield in the U.K. More complex examples related to health and perception can be found in [43,44]. In the first case, the authors used entropy-based concepts for knowledge discovery in heart rate variability, whereas in the second example, approximate entropy was used for EEG data. Finally, among the newest works from the medical domain, Coates et al. [45] used entropy in the Parkinson’s disease recognition process.

## 3. Methodology

For a set of objects *X*, every element can be described by a vector of *n* conditional attributes $\vec{x_{atr}}=\left\{x_{atr^{1}},x_{atr^{2}},\ldots ,x_{atr^{n}}\right\}$ where *n* is a number of conditional attributes. A decision class is denoted as $x_{class}$. Thus, every object is described by a pair $\left(\vec{a_{atr}},x_{class}\right)$. For every conditional attribute, we have the attribute and value pair, and every attribute can have a numeric or symbolic value. In the case of attributes with continuous values, the discretization procedure, leading to limiting the number of values for a single attribute, is often performed.

In classification problems, the decision class $x_{class}$, including information about the decision class for a single object, has one of the values belonging to the decision class set of values.

In this article, we perform the preprocessing of real-world data, which allows transforming the initial raw data into a decision table defined as follows:(1)$$DS=(X,\vec{x_{atr}},x_{class}).$$All analyzed data differ in terms of the size of set *X* and the number of attributes in the vector of conditional attributes $\vec{x_{atr}}$. We did not assume simplifications related to the cardinality of the decision class. Thus, for some sets, this attribute is continuous, and an additional discretization procedure is needed. Eventually, for all datasets, the number of values in decision class $x_{class}$ is discrete.

### 3.1. Entropy as a Measure of Classification Uncertainty

According to our aim, we wanted to explore the possibility of using entropy as an indicator of data difficulty. Therefore, we treated entropy as a measure of classification uncertainty. In addition, we explored how data can be simplified using only attributes selected in terms of entropy value. Therefore, we also examined the information attribute to assess the usefulness of entropy for data simplification.

Assuming that several different symbols describe information, entropy, in its basic form, can be calculated as follows:(2)$$E\left(DS\right)=-\sum_{i=1}^{\left|C\right|}p_{i}\cdot log\left(p_{i}\right),$$
where |C| is the number of different decision classess, and $p_{i}$ is the probability of occurence of the *i*-th decision class. With such a definition, entropy can be understood as a measure of data complexity. With an increasing number of decision classess available in the data, the overall complexity increases. In the most trivial case, for a single decision class, the $p_{i}$ value is equal to 1, whereas $log(p_{i})$ is zero (as well as the entropy). Thus, any increase in this value leads to higher entropy.

The value of the information attribute (Equation (Equation 3)) is determined for each conditional attribute to determine how it can change the entropy of the decision table DS. The resulting value determines the entropy that can be obtained by considering that attribute.

The information attribute is thus based on the calculation of entropy due to decision classes (Equation (Equation 2)), but this is performed due to the cases grouped by the values of the attribute being analyzed.

Formally, the information attribute is written as Equation (Equation 3), but note that these determinations are required for each attribute, where *k* is the number of attributes being analyzed, *m* is the number of possible values of the *k*-th attribute, and $|DS_{i}|$ is the number of instances having the *i*-th attribute value (analogously, $DS_{i}$ is the subset of the decision table DS that has only the *i*-th attribute value on attribute *k*).
(3)$$info\_att\left(k,DS\right)=\sum_{i=1}^{m}\frac{|DS_{i}|}{\left|DS\right|}\cdot E\left(DS_{i}\right)$$

In our considerations, info_att is crucial for simplifying the dataset. For each decision table DS with the number of conditional attributes *n*, values are determined based on Equation (Equation 4). This observation is used for further analysis.
(4)$$all\_info\_att\left(DS\right)=\sum_{k=1}^{n}info\_att(k,DS)$$

### 3.2. Classification Measures

In our research, we wanted to examine the classification quality using state-of-the-art machine learning algorithms. We chose decision trees (CART algorithm) and ensemble methods: Random Forest, Bagging, and AdaBoost. To assess the quality of classification, in addition to the classical measures of classification quality (accuracy), we also used precision (called positive predictive value (PPV)) and recall (called true positive rate (TPR)). Notably, these are binary classification measures, i.e., for a dataset with only two decision classes. In real datasets, there are often more decision classes. Several methods can be used to generalize precision and recall. We wanted to provide as much information as possible in our solutions, so we computed precision and recall for each decision class.

Therefore, for PPV, the analyzed decision class is treated as positive and all others as negative, and analogously for TPR. So, in the definition of the measures of the quality of classification (accuracy in Equation (Equation 5), precision in Equation (Equation 6), and recall in Equation (Equation 7)), we denote:**TP** **:**to identify all correctly classified cases of the analyzed class;**TN** **:**to identify all cases outside the analyzed class that were not assigned to this class;**FP** **:**to identify all cases outside the analyzed class that were assigned to this class;**FN** **:**to identify all misclassified cases of the analyzed class.
(5)$$accuracy=\frac{TP+TN}{TP+TN+FP+FN},$$
(6)$$PPV=\frac{TP}{TP+FP},$$
(7)$$TPR=\frac{TP}{TP+FN}.$$

## 4. Data Preparation and Preprocessing

In this section, we provide details of the real-world data used in further experiments. The data were collected from external sources and cover various fields. We adapted the raw data into a decision table format, described in detail in the previous section, to perform the tests based on the classification problem. All necessary steps for data processing are described in this section.

However, despite the processing of all datasets, some general preprocessing steps were used. Below we indicate these steps in points with a short description.

collect data in the raw format—the first step was to obtain the entire data. Please note that for some cases, these data were obtained from different sources; however, all information initially was presented as a table;join data tables from different sources—this step was used to merge all obtained data into a single table structure;eliminate all missing and incomplete data—no artificial methods allowing to repair missing data were included in this point;eliminate potential outliers in the data—by outlier, we mean observation outside the range 〈Q1−3·IQ:Q3+3·IQ〉 (where *Q*1 is the first quartile, *Q*3 is the third quartile, and *IQ* is the interquartile range);perform discretization for selected attributes (attributes pointed out by the domain expert having a relatively large number of values).

Please note that the last step was used for both conditional attributes as well as the decision attribute (if needed). Moreover, these were general steps adapted for all data. However, additional steps were explicitly performed for the selected data (for example, related to the natural language processing), described in detail in subsections related to different data.

### 4.1. Fake News Data

Universal access to the Internet created the possibility of the rapid creation and gaining of knowledge by users, which became a threat through the easy spread of false information in the form of fake news. Fake news aims to present users with a view that is not in line with reality or leads them to make wrong decisions or actions based on false information.

The problem of disinformation is best visible on social networking services and news sites, where fake news is spreading widely in the form of sharing, passing on to friends, or creating documents based on unreliable sources [46]. Therefore, it is essential to quickly classify the documents posted and adequately mark the articles as true or fake news. The subject matter of the documents from the fake news dataset is related to many different fields; in particular, it concerns political, media, and financial content, as well as current events [47,48].

Kannan et al. [49] claimed that preprocessing real text data for analysis using machine learning algorithms is always the longest stage and often amounts to around 80% of the total processing time. Therefore, to transform the fake news dataset into a decision table, we propose applying the statistical approach of natural language processing (NLP).

In the first step of NLP, the tokenization process is carried out, dividing a given text into the smallest unit (e.g., a sequence of words, bytes, syllables, or characters) called a token. The result is the creation of an n-gram model that is used to identify and analyze attributes used in natural language modeling and processing [50]. In our research, we used n-gram to define individual words from document titles, from which we additionally rejected words appearing on the stop word list. An example of a stop words list is presented in Figure 1.

The next step in NLP is to perform the normalization process using two methods: stemming and lemmatization. The stemming method is used to extract the subject and the endings of the words. Eventually, similar words are replaced by the same base word [51]. The method of lemmatization consists of reducing the word to its basic form [52]. The purpose of the normalization process is to reduce the variability in the set of terms.

The final step in the NLP covered in this research is creating a word vector model as a document representation. Our vector model is presented as a matrix (Figure 2), where documents ( dok_1–dok_n) are presented in the form of feature vectors representing particular attributes (at_1–at_n). In the model, we use a binary representation, where each value from the {0,1} set determines whether the word appears in a given document. In addition, the number of attributes is limited to the most common words in the title of the document. On this basis, the fake news dataset was transformed into a decision table consisting of the attributes of the most common words and a decision attribute (decision) containing two classes (true or fake).

The decision table structure consists of columns with conditional attributes and one decision, whereas rows include all documents from the set. Conditional attributes are words most often appearing in the text. The presence of specific words (in the decision table) is strictly dependent on the analyzed dataset. For this reason, the number of attributes is limited. Table 1 shows an example of the frequency of words (selected as conditional attributes) in the titles of true and fake news.

Real text datasets are challenging to analyze due to the large number of attributes [53] that constitute single words for the fake news dataset. The distribution of attribute values due to decision classes (fake and true) is presented in Figure 3.

For each attribute, there is one histogram (Figure 3) consisting of two columns, which corresponds to the number of values for each attribute. The first column shows the number of objects (article content) in which the selected word does not appear (as an attribute value), while the second column shows the number of objects in which the selected word appears at least once. These numbers are shown in the chart. Additionally, each column shows the assignment of a word to the appropriate class: blue is the true class, and red is the fake class.

By such a distribution of attributes due to decision classes (fake and true), it can be seen that some words (such as word_3, word_7, word_17, word_19) do not appear at all in the fake class—the right column is entirely blue. However, in the case of the first column, the division into both classes is equal for almost all attributes.

### 4.2. User Websites Navigation Data

Electronic commerce (e-commerce) has become popular as the Internet has grown, with many websites offering online sales, and e-commerce activity is undergoing a significant revolution. The major challenges in research are the collection, identification, and adoption of data supplied by Internet services to provide actionable marketing intelligence.

The main difficulty in web usage mining is the procurement of the desired database, as the only information we can collect from users visiting a website is through tracing the pages they have accessed.

Data collected from log files must be processed before data mining techniques (based on machine learning algorithms) can be used. Then, the personalization process is performed in the six main steps generally used in the field:**Data collection**: Collecting the data from the server or the user side.**Data filtering**: Removing or correcting undesirable data such as the log information obtained by crawlers.**User identification**: Identification of user by IP address, cookies, and direct identification.**Session identification**: Tracking the activity of the same user.**Characteristics selection**: Selecting characteristics that can be useful for user behavior analysis.**User behavior analysis**: Studying the behavior of different users for selecting dominant ones (i.e., the characteristics that change significantly from one behavior to another).

The main idea of analyzing the users’ behavior during user navigation was to limit the users’ sessions to 10 actions. Each action corresponds to a one-page view by the users. We chose the 10 actions limitation in the session because it was impossible to perform a pertinent clustering using less than 10 actions for the user session; the cluster was not significant enough, and differences between clusters were negligible.

Before the phase of navigation conditional attributes selection, the hierarchy of the website was derived. An example division of the site is as follows: First, we separated thematic websites to create universes. Websites from each universe were about the same topic. Then, we divided the entire site into seven different universes:**Store**: the main universe (for example with products list),**Quick order**: direct purchases by entering the catalog reference,**Shopping cart (purchase)**,**Sales**,**Consulting**: customers questions and FAQs,**Condition**: Terms of sale and shipping, and**Various**: all others, such as home pages.

The universe store was divided into three levels of hierarchy: section, subsection, and subsubsection. Generally, the final product page corresponds to the subsubsection.

From this hierarchy, we selected conditional attributes that describe the user navigation of our commercial partner’s website. The attributes are presented in Table 2.

The presented attributes are described as follows:**User ID** describes the ID based on cookies, a unique ID for each user;**Session ID** describes the session ID during one day each session is considered as closed after 30 min of inaction;**Purchase** is a binary value that shows if the user made a purchase during their action;**Discount code** is a boolean value describing the presence of a discount code during the purchase;**New user** describes whether the user was recognized as a user who already made a purchase on the site;**Source of navigation** describes whether the user is entered into our commercial partner’s site voluntarily by using, for example, the search engine, or was pushed to visit the site by a mail company;**Total time** describes the length of a session in seconds;**Total universe (1–7)** represents the seven different attributes that describe the time that a visitor spends in each universe;**Total no. of pages seen** describes the number of all the pages visited by the user during a session;**No. pages universe (1–7) seen** represents the seven different attributes that describe the number of pages visited by a user in each universe;**No. of universe, section, subsection, and subsubsection changes** are the four features that describe the number of changes the user makes during their navigation. If, for example, the user switches universe and then returns to the previous one, the value of this attribute is equal to 2;**No. of sections or subsections seen** are the two attributes that describe the number of different sections or subsections seen during the user session;**No. of product pages seen** describes the number of product pages seen in total;**No. of same product seen** describes the sum of product pages that have been seen several times.

For the decision attribute, we chose the binary attribute purchase. Decisions classes were “yes” and “no”. All the attributes were normalized.

The distribution of attribute values due to decision classes—purchase is presented in the Figure 4. Two colors correspond to the decision classes: blue indicates sessions not completed with the purchase, and red indicates the sections in which the purchase was made.

As we can see in the Figure 4, some attributes do not discriminate the decision class. For example, the decision class distribution is identical for attributes such as Day/Month/Year, Hour_of_end, and Source_of_navigation. On the other hand, attributes such as Discount_code, Total_time, or New_user, clearly indicate the purchase class. According to the presented data distribution, we can determine that the user’s session ending with purchase has the following attribute values: avg._no_of_pages_viewed and average_amount_of_time_spent_on_navigation, the customer is not the first time on the website, and he has a discount code, the customer does not spend a lot of time in the store section, but frequently changes subpages in this category.

### 4.3. Real Estate Market Data

The real estate market has grown rapidly during the recent years [54]. As such, both the volume of data and the number of processed details have increased. Investors are looking for attractive properties from which profit can easily be earned. As customer habits change, so do the features connected to a particular property that is essential for buyers.

The change in investor and end-consumer behavior has led to the inclusions of additional details in advertisements of properties. Each advertisement is currently filled with much additional information, some of it structured and some of it only provided in descriptive text. The real estate market data used in this paper originated from actual advertisements presented on multiple Polish market web pages. The details of the adverts are often hidden inside the text describing a particular property. However, many details are often presented in a structured form, allowing less sophisticated automatic scrapers to gather the data. For some of the conditional attributes, it is still necessary to perform more advanced processing. For instance, the_floor_number is usually provided as a number in the vast majority of cases. However, there are some occasions where it is stated verbally as “ground floor” or “higher than the 10th floor”. Most of the advertising portals do not provide a good enough validation of this data, which is why, during the data acquisition, we had to construct more detailed methods to handle the special types of values and data. A similar process had to be performed for geo-encoding the spatial data. In almost every advertisement, the exact address of the property was not given; only the street name and the city were described. Sometimes the street names had spelling errors, were not correctly placed on a map, or used an old street name before the mandatory change of street names in Poland that recently occurred [55].

Notably, the process of acquiring data from web pages is complicated. The dataset used in the current study consists of the following conditional attributes:**Build date** is the year the property was built. This attribute needed extra preprocessing steps, as some of the records provided a textual representation such as “the late 1980s”. Therefore, the dates were given as is without any numerical processing.**Total number of floors** in a whole building. As mentioned before, more advanced NLP methods were applied to clean up the data.**Building material**. As the materials change across the decades, a whole dictionary of construction materials was created using both automatic methods and expert knowledge. We also constructed a synonyms dictionary. The provided value was then compared to the dictionaries and cleaned up. This is a categorical attribute.**Floor number** on which the particular property is situated.Area] of the property. Here, the vast majority of data were provided in square meters, but some of the land properties provided this value in acres, which had to be converted to an SI-derived unit of measure.**Building type** is a categorical attribute denoting the building type (e.g., semi-detached building, loft, etc.). Here, we used similar preprocessing techniques to those used for building material.**Condition state** describes the overall condition of a property. As this is highly subjective, as there are virtually no norms that can standardize this attribute, we used a two-fold approach. As a starting point, the value presented in the advertisement was taken directly as-is. Next, this value was then compared to the dictionary of values and corrected for spelling errors and synonyms. In a second step, the description text was analyzed to find keywords that could decrease or increase the overall condition of a property. For instance, if the property was marked as “ready to move in”, but the description mentioned that “painting needed” or “kitchen is not equipped with stove”, the overall condition was decreased. Although this is considered a categorical attribute, current works involve introducing the order relation to items from the condition dictionary. Additionally, we are working on an image classifier that will automatically label the state of a property.**Windows** with which the property is equipped (wooden, PCV, etc.).**Private ad** is a dichotomous attribute discriminating if an advert was published by a professional dealer or a private party. As research has shown, these two types are constructed vastly differently. Most of the time, private advertisements have lower-quality photographs, but the description is more accurate and meaningful than in professional ads. The former often includes additional costs in the description (such as a mandatory extra-paid parking space).**Market type** has two values: primary and aftermarket.**Ownership type** describes the legal ownership type of a given property.

The last attribute, being the decision one, denotes the price per square meter. As this value can fluctuate widely, we transformed it using a simple discretization: (8)$$bucket=\frac{price\_per\_sq\_mt}{1000}.$$

Because of the nature of scrapped data and the frequent necessity for repairing or transforming the data (e.g., converting units of measurement between imperial and metrical), this data is rather difficult to analyze. Furthermore, many attributes, all interesting for the end-user, make this processing even more complicated.

The distribution of attribute values in accordance with decision classes was created, as shown in the Figure 5. Please note that due to many values in the decision class, there was no visible distinction related to color for each class.

Even though the data has been preprocessed extensively, some of the original values with mistakes were left intact. This is the case for area attribute, where one of the flat’s areas is set to 349,000 square meters. This is clearly seen in the distribution plot, where the plot is heavily skewed. The same thing is happening with the build_date (a building has a date set to 892,007; there are also some spelling errors with a date like 19,000 or 20,014 where an individual probably inserted an additional 0). Because the number of records with such mistakes is relatively small (less than 0.02%), the authors included these outliers in the dataset to determine their influence on the overall entropy and classification results.

It is clearly seen that most of the properties are situated below fourth floor, which is expected, as it is far more easy to build such buildings in Poland compared to skyscrappers due to legal reasons. The owners tend to over-estimate the quality of interior, therefore the vast majority of apartments have the “ready to be moved” condition_state. Most of the analyzed apartments also have modern PVC windows.

### 4.4. Sport Data

Sport is a valuable part of many people’s lives, understood both as physical activity and in terms following individual teams or athletes. Football is the most popular sport known, with the European leagues being some of the most famous in the world. Therefore, the top leagues from Germany, Italy, and Spain were selected for our analysis.

Numerous studies based on both expert analysis and machine learning techniques for predicting sports results can be found in the literature [56,57,58,59]. The most popular and accessible are predictions of match results in the form of win/loss/draw; however, both analyses and predictions may concern other elements such as the number of goals scored, the exact score, or the number of yellow cards [56,60].

The dataset was created from the tabular data available on a website [61]. For complete information, the data were extracted using the scraping method from two tables. The first one contains data about the league table. The second one consists of information about individual matches. The tables were then combined to obtain a full decision table that was divided into sets for each country. The conditional attributes included in the decision tables are presented below:**Season**: The season in which the games were played: a nominal variable using data for 10 consecutive seasons from 2011–2012 (“11/12”) to 2020–2021 (“20/21”).**Round**: The number of competition rounds. A quotient, integer variable ranging from six to 34 for Germany, and to 38 for Spain and Italy. Based on conducted experiments and the arguments indicated in the literature, the data for the first five rounds of each of the seasons were excluded from the analysis [62].**Team1**: The name of the first team. Categorical variable taking different values 28 for Germany 28, 34 for Italy 34, and 33 for Spain.**Position T1**: Position of Team1 in the competition table. A quotient, integer variable ranging from 1 to 18 for Germany, and 20 for Spain and Italy.**Match T1**: Match played by Team1 up to the current round. A quotient, integer variable ranging from six to 34 for Germany, and to 38 for Spain and Italy.**Winnings T1**: The number of matches won by Team1 up to the current round. A quotient, integer variable ranging from six to 34 for Germany, and to 38 for Spain and Italy.**Draws T1**: The number of draws for Team1 up to the current round. A quotient, integer variable ranging from six to 34 for Germany, and to 38 for Spain and Italy.**Losses T1**: The number of matches lost by Team1 up to the current round. A quotient, integer variable with values ranging from six to 34 for Germany, and to 38 for Spain and Italy.**Goals scored T1**: Goals scored by Team1 up to the current round. A quotient, integer variable.**Goals conceded T1**: Goals conceded by Team1 up to the current round. A quotient, integer variable.**Goal difference T1**: Difference between goals scored and lost by Team1. A quotient, integer variable.**Points T1**: The number of points gained by Team1 up to the current round.**Series T1**: Series of results match for Team1. A nominal variable, consisting of three symbols containing information about the results of the last three games played by the team. In the first position, there is the last played game, where W is team wins, R is a draw, P is team loss, and B indicates no data.

The same attributes are available for the second team as for the first team. The conditional attributes for the second team were marked by “T2”. The last of the attributes is the decision class (match_result), which can have three values: 1 indicates a win for team 1, 2 indicates a win for team 2, and X is a draw. Team 1 is the team playing the game on its home field; team 2 is the team playing away.

The Figure 6 shows examples of distributions for the data of the German Bundesliga. A significant part of the data is characterized by right-hand asymmetry, which is naturally related to the domain specificity of the data. Representative examples of this fact are, among others, Winnings_T1, Draws_T1, Goals_scored_T1, Goals_conceded_T1, Points_T1. A team starts with a value of 0 for the number of games won/lost, goals scored/conceded or the number of points. During the game, teams increase the values of these attributes, or they remain unchanged. This behavior contributes to the right asymmetry in the data. The distribution for Goal_difference_T1 is much closer to the normal distribution. In the decision class distribution, it can be seen that the most common values are related to the home team win (color = red), then the visiting team wins (color = cyan) and draw (color = blue). The last two classes have numbers much more similar to each other. The following rules are also observed for the “Team2” data and for other countries’ leagues.

### 4.5. Financial Data

From financial data, we can highlight two main groups of data. The first one is related to the well-known Markowitz model (and its extensions) and the portfolio selection problem, which is beyond the scope of this study. The second group is related to the price and indicator data from various markets. In this group, the most popular data are obtained from the financial markets (also known as forex market or foreign exchange market) and concerns the currency pairs.

A single market indicator (or group of indicators used jointly) is used in trading systems to generate buy signals. All indicator data were calculated according to market indicator formulas, which can be divided into two separate groups. The first covers trend-following indicators, which include the moving average (MA) market indicator. The MA for time *t* and *s* periods, denoted $MA_{s}\left(t\right)$, is calculated as:(9)$$MA_{s}\left(t\right)=\frac{\sum_{i=t-s}^{t-1}price_{i}}{s},$$
where $price_{i}$ is the value of the corresponding instrument at time *i*. In the above context, the period is the number of values considered when calculating the indicator. The second group of indicators covers the oscillators, whose primary purpose is to indicate rising or falling potential for the given currency pair. The indicator value is calculated using the currency value and can include the closing, opening, minimum, or maximum currency pair value from previous sessions (or any combination of the above). As an example, the oscillator Relative Strength Index (RSI ) is calculated based on the last *n* periods in time *t* as follows:(10)$$RSI_{s}\left(t\right)=100-\frac{100}{1-\frac{avg_{gain}}{avg_{loss}}},$$
where $avg_{gain}$ is the sum of gains over the past *s* periods and $avg_{loss}$ is the sum of losses over the past *p* periods.

All mentioned, indicators are calculated based on the currency pair value, which was included in the data. The decision (BUY or SELL) is based on the indicator value in time *t* and its relation to the indicator value at time t−1. Therefore, the general rule for opening the trade for indicators can be defined as follows:(11)$$cond_{Buy}=true\mathrm{if}\left(ind_{s}\left(t-1\right)<c\right)\wedge \left(ind_{s}\left(t\right)>c\right),$$
where $ind_{s}\left(t\right)$ is the value of indicator ind in the present reading *t* considering the last *s* readings, t−1 is the value in the previous reading, and *c* is the indicator level (different for each indicator), which should be crossed, to observe the signal.

As shown, the crucial aspect related to generating the signal by the indicator is the value difference between two successive readings. Thus, we decided to include this information in our data in some limited way (in the case of the MA indicator). For the remaining indicators, a discretization procedure was performed because, in the classification process performed in the experimental section, only a limited number of indicator values was accepted. The summary for each indicator is presented in Table 3.

Each of our readings in data also included the decision taken as one of the following values: STRONGBUY, BUY, WAIT, SELL, or STRONGSELL. Each set’s decision was based on calculating the difference between the present instrument value and the value observed after *p* readings. This schema is presented in Figure 7. In this study, we examined *p* equal to 5.

The distribution of attribute values in accordance to decision classes was created, as shown in Figure 8. We selected an example data for the AUDUSD instrument; however, a similar distribution of attribute values was noted for the remaining datasets. The blue color on the chart denotes the number of objects for which the STRONG BUY class was observed. Cyan color is related to the STRONG SELL class. Both classes cover the majority of all objects in the data. The red color shows the objects belonging to the SELL class. The two remaining classes are BUY and WAIT, respectively.

In general, we can divide the whole attribute set into three different categories. The first one is related to the instrument price (which is Close on the chart) and two indicators (the_moving_average) based on the price. For this category, we observe attributes, for which there are several values with a reasonably high number of objects assigned. The second category is related to the same indicators, where the difference between two successive readings was calculated. It gives us a distribution close to the normal distribution, where the minor differences (close to the 0) have a high number of objects assigned. Finally, the last category is related to the oscillator indicators like Bulls or OSMA, for which once again the approximation of the normal distribution is observed. Also, for these attributes, relative change between successive readings was included. The main problem in this data is that the slight differences (the middle part of attributes number 4 to 11) are frequently observed in the data. At the same time, most information comes from the relatively significant differences (tails of the distribution). Thus the most promising attribute values are the least observed in the data.

## 5. Numerical Experiments

In this section, we describe the experiments we performed on different real-world datasets. For every set, the experiments consisted of four steps:calculation of the information for each conditional attribute (information attribute);classification of the obtained data;classification on the limited set of attributes (including the best 25% of the conditional attributes selected based on the information attributes) as well as the classification on the set of attributes selected by the correlation-based approach;sensitivity analysis on the parameter related to the percent of attributes included in the limited set of attributes.

We selected a group of well-known state-of-the-art algorithms for the classification: decision tree, Random Forest, Bagging, and AdaBoost. Two measures were used to estimate the quality of classification: the positive predictive value (PPV) and the true positive rate (TPR). Additionally, the accuracy of the classification was measured.

### 5.1. Fake News Data

The fake news detection research was conducted on the ISOT Fake News Dataset provided by the University of Victoria, Canada [63]. This collection includes 44,898 documents, of which 21,417 are real news cases and 23,481 are fake news. Each document in the set is described with the following attributes:title,text,subject,date.

Additionally, to determine the decision class, the main file was divided into two separate files:FAKE: documents that were detected and marked by Politifact.com as untrue sources;TRUE: real documents from Reuters.com, accessed on 31 August 2021.

In our fake news detection experiments, the dataset was limited to the title, and the *decision (true or fake news)* attributes only. This restriction allowed us to quickly mark the document based on the title without analyzing its content. In our previous research [64], we showed that the fake news detection model analyzing the titles produces accurate results and reduces the runtime of classification algorithms compared to the analysis of the entire content of the document.

In the first step of the experiments, we calculated the entropy of the decision class (see Equation (Equation 2)) and the information for each conditional attribute, which were the most common words in the documents. Notably, the values of the decision table (frequency of the occurrence of certain words) are strictly dependent on the documents that comprise the set on which the algorithm was trained. For this reason, the number of attributes was limited to 20. The results of this experiment are presented in Table 4. As can be seen, almost all information values for individual attributes are close to the maximum entropy value (1.0) and are in the range of 0.958–0.998. However, the last row in Table 4 shows the entropy value for the entire dataset.

In general, it is difficult to determine the set of attributes that most impact the classification results. Only attribute *word_17* has an advantage over other attributes because, for attribute *word_17*, the value of the information is visibly lower and amounts to 0.83. This means that after a single attribute—in this case, one word per document title—whether the document’s full title is true or false cannot be determined. Moreover, the conditional attributes are different for a different set of documents, which entails the possibility of entirely different entropy values.

In the next step of the experiments, the values of the classification evaluation measures were calculated using selected machine learning algorithms, which were derived for each of two decision classes (true or fake news). Table 5 shows the results for the classification of fake news data by decision class for all twenty attributes.

In the case of decision class FAKE, PPV values were in the range of 91.38–98.88%, where the best result was obtained using a decision tree, where TPR values were in the range 46.05–58.67%, and the best result was obtained with Bagging. However, in the case of decision class TRUE, PPV values were in the range of 62.70–67.46% (Bagging was superior), and TPR values were in the range of 94.65–99.43% and the best results were obtained by the decision tree.

We also checked the influence of a limited number of attributes on the classification results. For this purpose, 25% of the attributes with the lowest value information attribute were selected (in this case, the top five attributes were selected). The obtained results are presented in Table 6, where the values are similar to those in Table 5. This proves that with a significantly limited number of attributes—in this case, up to 5 single words per document—the classification results for the algorithms used are the same as for a full set of conditional attributes.

The classification accuracy values for the entire set were calculated in terms of the number of attributes (five or 20 attributes), and the results are presented in Table 7. As can be seen, for the three algorithms, the accuracy was in the range of 74.17–75.49%, while for the decision tree, the accuracy was slightly lower at 71.51%.

When detecting fake news by title only, the classification accuracy measure determined how many documents were correctly classified. However, when using the PPV and TPR measures, it was possible to assess how many documents in a given class were correctly recalled and with what confidence (precision).

### 5.2. User Websites Navigation Data

The vital part of preprocessing the data is converting the raw data into a set of navigation attributes. During our research, we obtained the data of our commercial partner for one entire year. This data was more than 85 GB in size. For our learning base, we used a sample of data of one month. We chose the month of April due to avoid any marketing actions. The database for one month represents more than one million sessions with more than 10 actions performed. On account of the scale of the database, the treatment is time-consuming. After performing the limitation, we obtained 211,639 user sessions.

For entropy and classification analyses, we eliminated significantly correlated attributes such as total_amount. In the end, we obtained 31 attributes and one binary decision attribute, purchase.

The dataset for user behavior analysis consists of 211,639 unique rows. Each entry represents a unique user navigation session. First, the entropy value represents the entropy of a decision class of individual conditional attributes. Second, the results are shown in Table 8 along with the cardinality of the value set for each conditional attribute.

The entropy values for most attributes were near 0.5. For several attributes, the entropy value was lower than 0.5. For few attributes, the entropy was less than 0.2. An explanation may be the distribution of values for these attributes, which was strongly unbalanced. In most cases, the value of an attribute was equal to zero, only occasionally taking different values. Examples of these attributes are discount_code and new_user. When analyzing other attributes, the values of entropy were similar, indicating that most attributes carried an equivalent level of information. Intuitively, it seems that some attributes should be more discriminatory, but the analysis of the results did not confirm this. There were no highly biased attributes in the analyzed dataset.

Table 9 provides the classification results for the same dataset divided by each decision class value. The efficiency measures indicated relatively accurate results: PPV, TPR, and accuracy values were in the range of 0.89–1. However, both PPV and TPR were better for the decision class equal to “no”. The results for the decision tree, Random Forest, and AdaBoost were similar. The results obtained using the Bagging algorithm were visibly worse than for the other algorithms. The PPV value for the class “yes” was around 0.5. Again, the reason seems to be the uneven distribution of the values of the target class.

Finally, we performed the limitation of the attributes used in classification. The limitation was based on the analysis of the value of entropy for each attribute. We selected the 25% most significant conditional attributes and performed the classification with a limited number of attributes. The classification results for user websites navigation data by decision class values for 25% of the attributes with the lowest information attribute are presented in Table 10.

The accuracy of the results for user websites navigation data is compared in Table 11. The number of all attributes participating in the classification process was 31. After limiting the set of attributes to seven, the results of the classifier efficiency increased, which may be counterintuitive. Depending on the classifier used, the improvement in efficiency ranges from 0% (DT) to 10% (Bagging). The presented analysis shows the importance of limiting the attributes at the data preprocessing stage and of classification parameterization.

### 5.3. Real Estate Market Data

The goal of the real estate market data experiment presented in this paper was to find which attributes are crucial and essential for AI model creation based on the presented decision table. To achieve this, the values of the information attributes were computed.

The dataset consisted of 14,344 unique rows. There were 13 conditional attributes (described earlier) and one decision (price bucket). In the first experiment, we computed the entropy of a decision class and the information of individual conditional attributes. The results are shown in Table 12, along with the cardinality of the value set for each attribute.

Because the data were obtained from actual advertisements, the cardinality of a decision class fell more or less in a normal distribution (Figure 9). The most frequent price fell into the PLN 6000–7000 per square meter bucket. The far-right side of the histogram plot shows the luxury properties that are part of the dataset. Remember that the property’s region heavily influences the real estate market. A property located in the capital is far more expensive than the same property in a less rich part of the country. The overall decision entropy is relatively high, as the classification problem is rather difficult. Most of the attributes maintain a similar entropy value, with a single exception being the property area. Because of the cardinality of this attribute and the fact that the price of a property is usually heavily correlated with the location, this is to be expected. However, the surprising finding is that the value of entropy is also relatively high, which means that the price fluctuation between a property with a similar area is also significant. We found no noticeable changes in information for attributes such as market_type or ownership_type, indicating that such features have secondary importance for the selling price.

All attributes except the area obtained an information value close to the maximal entropy for the whole dataset. That means that no single conditional attribute was enough to predict the price bucket of a given property. Even the area conditional attribute, with a visibly lower information value equal to 2.07, was insufficient to correctly predict the price range. The price range agrees with intuition: a large but poorly located and unfurnished ruin might be cheaper than a downtown loft.

Table 13 provides the classification results for the same dataset divided by each decision class value. The Bagging algorithm produced the best results by far in nearly every decision class, both in terms of PPV and TPR. When using the limited set of attributes the following results were obtained (Table 14). Overall accuracy results were also superior using the Bagging algorithm (Table 15). Further research is required to determine whether a precise fine-tuning of hyper-parameters would increase the quality of results produced by the other algorithms.

### 5.4. Sport Data

Three datasets with 3362 unique rows for Spain, 2674 for Germany, and 3359 for Italy were analyzed. There was a total of 26 conditional attributes with match_result as a decision class. In the first stage, we calculated the entropy of a decision class and the information attribute values. The results are shown in Table 16, along with the cardinality of the value set for each attribute.

In all three analyzed datasets, the information attribute value was relatively small. It was the lowest for *Goal difference T1* and *Goal difference T2*, oscillating between 1.38 and 1.40. The highest information attribute value was recorded for *Season*. The next conditional attributes with high values were *Round* and *Matches T1 (T2)*. For the remaining measures, the values of attributes were similar. Table 16 presents the entropy of the datasets, all of which are similar (1.52–1.55).

Of the selected methods, random forest had the highest accuracy, followed by the AdaBoost algorithm. The decision tree performed the worst in the classification. None of the algorithms provided a significant advantage in terms of efficiency measures. A summary of the results is presented in Table 17.

Tests were also conducted using fewer attributes (from 24 to 6; 25% of the set based on the information attributes values). The results obtained are presented in Table 18 and Table 19. As can be observed, similar results were obtained with a limited list of attributes. For some cases, the results obtained with a limited set of attributes were better. The best algorithms, in this case, were AdaBoost and random forest, whereas Bagging worked poorly.

The results (Table 17) show a problem with the prediction of class X (draw), which is best exemplified by the complete lack of prediction results by the Random Forest algorithm for data from Germany and Spain; for the remaining cases, this class had poor results. The unbalanced values in the decision class may be the reason for this finding. Note that a draw between teams seldom occurs.

The classification accuracy for the three sets and all selected algorithms oscillated between 51.55% and 55.85%, being higher than the random approach (for the three decision classes = 33.33%). The Random Forest algorithm achieved the highest classification accuracy on the Italy dataset and the lowest was achieved by Bagging on the Spain dataset. The exact results are presented in Table 18.

### 5.5. Financial Data Results

We used daily forex data in this study, which means that every new value was obtained at the beginning of the daily market session. We selected four different currency pairs as separate datasets: AUDUSD, EURUSD, GBPUSD, and NZDUSD, each containing 2865 readings. In addition, we used six different oscillator indicators: the Bulls indicator (Bulls), Commodity Channel Index (CCI), DeMarker indicator (DM), Oscillator of Moving Average (OSMA), Relative Strength Index (RSI), and the stochastic_oscillator. Additionally, the moving average (MA) indicator, calculated for 14 (MA14) and 50 (MA50) past readings, were included. For the results, we used the MA indicator and MA to denote the absolute difference between two successive readings for the indicator. It provided us with an overall number for 10 attributes.

In Table 20, we present the entropy of the decision class along with the information attributes values for the four different datasets. Firstly, there are no visible differences between the entropy values for the different datasets. However, a significant difference exists in the case of trend-following indicators (the first four attributes related to the MA indicator). This is obvious for MA14 and MA50. However, these attributes were not preprocessed and were used as was. Small entropy values suggest the strong predictive power of these indicators; however, their practical usability is lower due to a large number of different attribute values (in comparison to other oscillator indicators such as RSI).

In the case of oscillators, information attribute values were held on the same level instead, and it would not be easy to identify the best (in the sense of information) indicators. However, it is easy to find many examples of articles confirming that indicators’ predictive capabilities are similar.

Table 21 presents the results of classification based on the PPV and TPR measures for the complete set of attributes available in the dataset. The decision class values were highly unbalanced, and for some cases, values such as BUY or SELL did not occur even once. For other cases (such as in the case of the GBPUSD dataset), the results were poor quality because we observed the STRONG BUY or STRONG SELL decision for most cases. However, in general, the AdaBoost algorithm for these rare cases with buying or selling values was slightly better than the Bagging algorithm. For the remaining cases, all four algorithms achieved similar results oscillating between 30% and 40%. Lower results for some cases (such as the STRONG BUY for the EURUSD dataset) could be related to the market situation and overall advantage of the bearish trend.

Next, we performed the classification once again on the limited set of attributes. The results are presented in Table 22. For both measures (PPV and TPR), the quality of classification slightly worsened. However, the results improved for some rare cases (for example, EURUSD and GBPUSD and the TPR measure). This was achieved despite considerably reducing the number of conditional attributes included in the classification process.

Eventually, we analyzed the classical accuracy measure for two cases: with the full set of conditional attributes along with the limited set. These results are presented in Table 23. Surprisingly, the results do not indicate that the full set of attributes allows obtaining the highest accuracy values. These results are ambiguous; for some cases, (AUDUSD or EURUSD with the Bagging algorithm), accuracy was higher using the limited number of attributes.

These observations were also confirmed for the remaining sets. Thus, it can be assumed that some core sets of attributes can allow obtaining a relatively accurate classification. However, dependencies between these attributes are more sophisticated than simple linear correlations.

### 5.6. Attributes Selection and the Sensitivity Analysis

To test and evaluate our results based on the attributes selection (based on the entropy values), we used the well-known correlation-based feature selection (CFS) method implemented in the WEKA system [65]. As a result, a subset of attributes, including the essential elements, were selected—comparison of a number of attributes obtained by our method and the WEKA system can be found in Table 24. As it can be noted, for most cases, the number of attributes in our approach is smaller than the number of attributes selected by the CFS method. For example, only the User Websites Navigation Data attribute selection is shown five instead of seven (out of 31 possible) attributes. In the case of the financial data, the number of attributes was the same for both methods. In contrast, for the remaining datasets, our proposed method allowed us to use a smaller number of attributes—extreme cases related to Real Estate Market Data indicated nine instead of three (out of 31) attributes.

A smaller number of attributes resulting from the use of our method does not affect the overall quality of classification. The results of classification after the selection are presented in Table 25 (names of datasets were written as an acronym). The table shows the difference in classification based on the attribute set calculated using the CFS method and our proposed approach. As can be observed, despite the smaller number of attributes indicated by the proposed method, the classification quality is similar—mostly does not exceed 0.3%. Only for the Random Forest method used for the Real Estate Market Data, an overall improvement close to 1% is observed—it is the case, where the number of attributes selected by the CFS method was equal to nine (instead of three in our proposed method). Similarly for the Sport Data, where there is improvement around 1%. While for the Financial Data, the highest differences (favoring our proposed method) were observed. In the case of the Random Forest and Bagging algorithms, the attributes selection worsens the results for over 2%. For the Financial Data for both cases, the classification was performed based on two attributes.

In the case of the proposed method, we used the threshold of 25% of attributes included in the classification. It was shown to evaluate if the small subset of attributes allows maintaining the relatively high classification quality. Attributes were selected as the most important from the point of view of the entropy measure. This threshold was set experimentally, and it was based on several different indicators. Going below the 25% could limit the subset of attributes to two or even a single value in the case of analyzed data. At the same time, in the case of many attributes, it was possible to observe the visible decrease of classification quality. An example chart for the Sport Data (Germany) is presented in Figure 10, where the quality of classification (the Y-axis) is presented depending on the number of attributes (the X-axis). The vertical line points out the 25% of attributes used in the article.

## 6. Conclusions and Future Works

In this study, we investigated the possibilities of using the entropy measure to select the best set of conditional attributes to be used in a classification problem. The general idea of the entropy, related works, and the problem background was introduced in the first part of the article. We also selected real-world data covering different fields. These data were retrieved and described with the use of domain knowledge experts. Finally, preprocessing was applied to all datasets, which were transformed into decision tables.

The datasets differed in their complexity, number of objects, number of conditional attributes, and the number of decision classes. Our goal was to calculate the entropy of decision classes and the information attribute values. Furthermore, we performed the classification with a set of well-known state-of-the-art algorithms. To estimate the quality of classification, we used the recall, precision, and accuracy measures. After the initial results, we selected the 25% best attributes (attributes with the best information attribute values) and performed the classification on the limited number of attributes.

For most of the cases, the algorithms obtained similar results. However, there were some examples, such as the real estate dataset, in which the Random Forest produced better results using only the limited attribute set. The Bagging algorithm showed slightly lower classification accuracy. The nature of the Random Forest algorithm, as the name implies, conducts each run providing similar but different results. The hyperparameters of Random Forest are the most prone to fine-tuning, but optimizing the parameter of each used algorithm for each used dataset was beyond the scope of this study. Notably, the value of real estate cannot be classified only using the significance of attributes but also must consider emotions and non-technical factors. For instance, we were unable to quantize the “cool” factor of a given property.

For the remaining datasets, the results were not uniform. It was difficult to identify the attributes with the best information attributes value. Differences in these values amongst the attributes in the single dataset were often negligible. However, eventually, we were able to select a subset of attributes with which the classification procedure was performed once again. Surprisingly, the limited set of attributes often allowed obtaining similar classification results. Unfortunately, it was impossible to capture the complex, nonlinear relations amongst the conditional attributes within the single dataset.

In the case of classification, we used the classical algorithms considered as a state-of-art approach. However, the multicriteria efficiency measure based on different entropy types could give much more useful information. This can be the case, especially for complex datasets without uniform structure (like Big Data). At the same time, we only investigated entropy in its basic form. An interesting approach could be related to introducing different entropy measures or even deriving estimates based on other entropy types.

In this article, we obtained some advantages over classical methods; however, the obtained results are not uniform. Therefore, our future goal could be related to extending the number of analyzed sets and emphasizing the quantitative results rather than focusing on the description of every single piece of data used in the experiments.

## Figures and Tables

**Figure 1 entropy-23-01621-f001:**
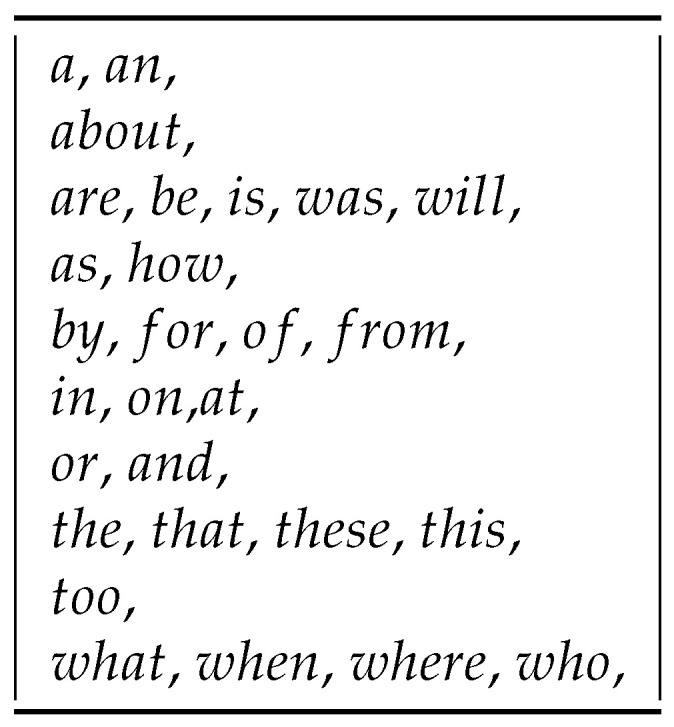
A sample list of rejected words, the so-called Stop Words.

**Figure 2 entropy-23-01621-f002:**
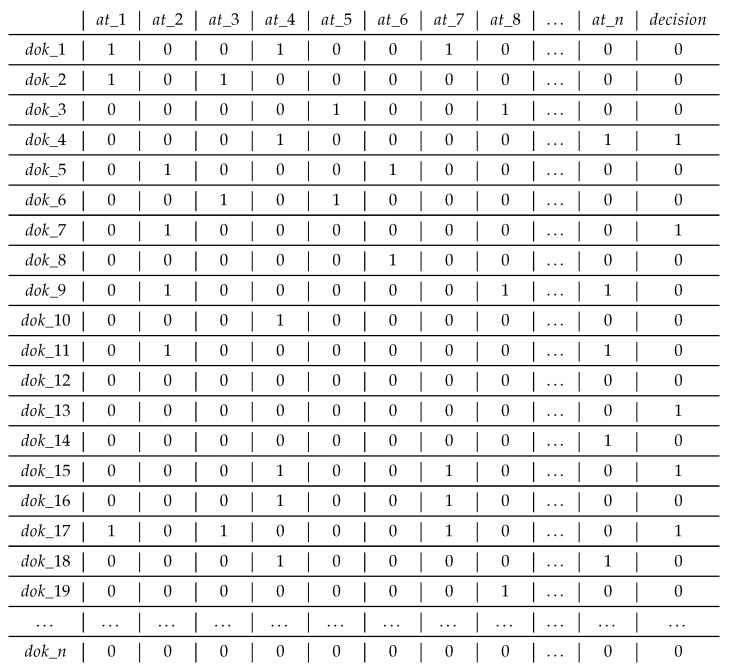
The sample matrix of words occurrence (selected as conditional attributes) in documents.

**Figure 3 entropy-23-01621-f003:**
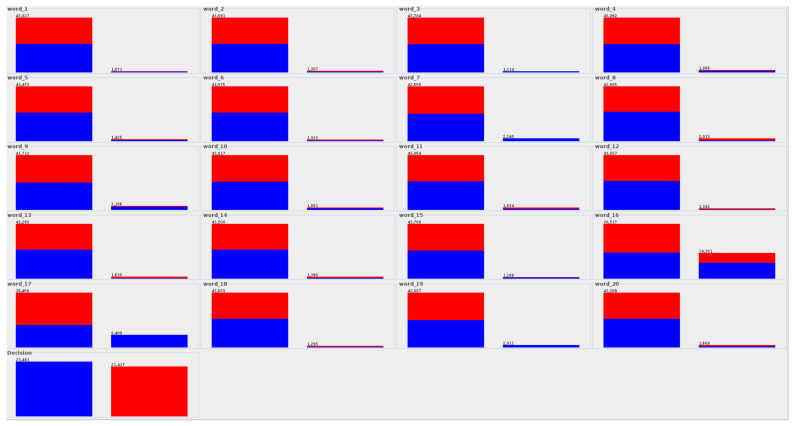
The distribution of attribute values due to decision classes for fake news data.

**Figure 4 entropy-23-01621-f004:**
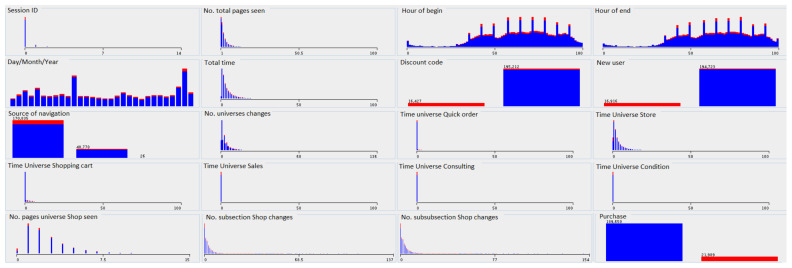
The distribution of attribute values due to decision classes for user websites navigation data.

**Figure 5 entropy-23-01621-f005:**
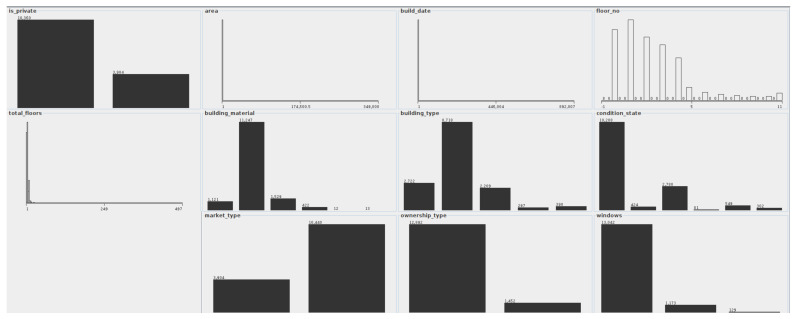
The distribution of attribute values due to decision classes for real estate market data.

**Figure 6 entropy-23-01621-f006:**
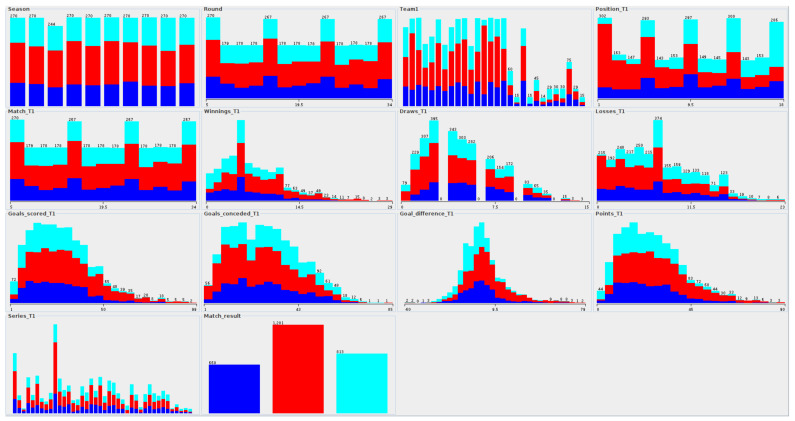
The distribution of attribute values due to decision classes for sport data.

**Figure 7 entropy-23-01621-f007:**
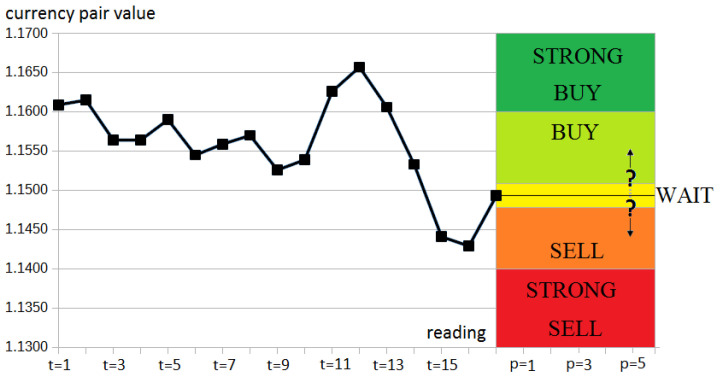
Decision calculation method for the financial data.

**Figure 8 entropy-23-01621-f008:**
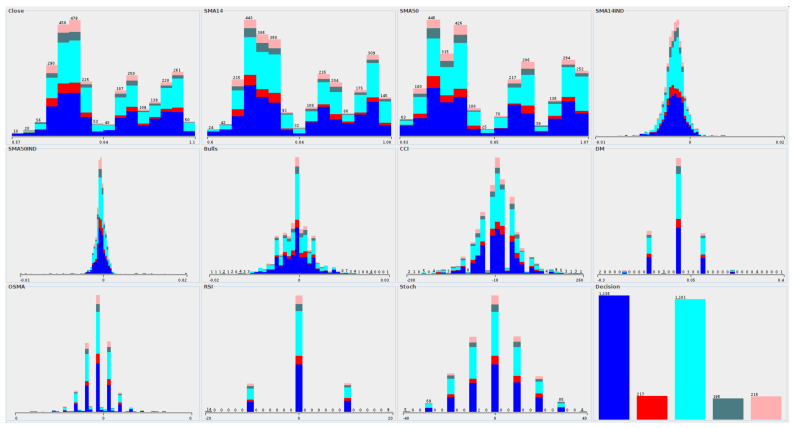
The distribution of attribute values due to decision classes for financial data.

**Figure 9 entropy-23-01621-f009:**
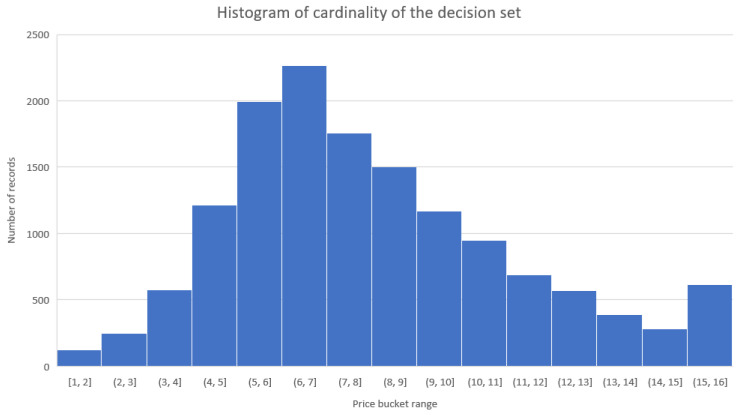
Histogram of cardinality of the decision set.

**Figure 10 entropy-23-01621-f010:**
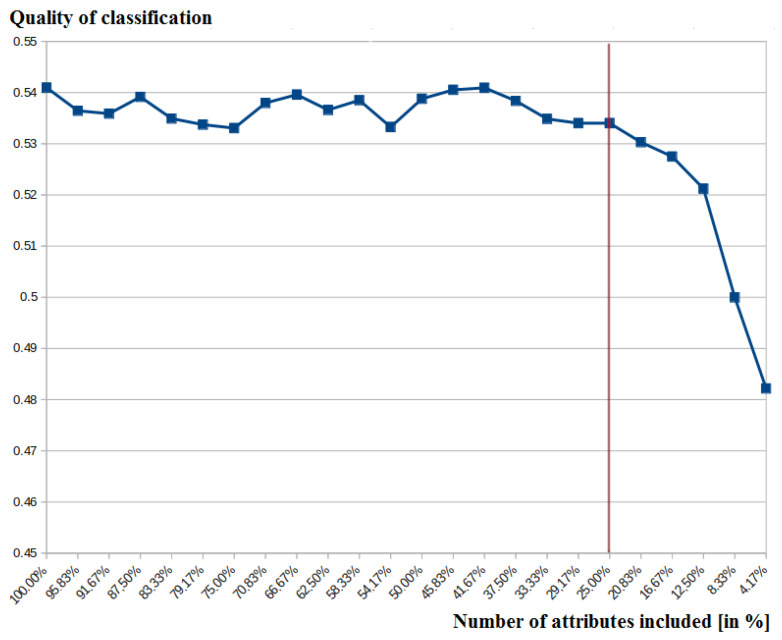
Classification accuracy depending on the number of attributes.

**Table 1 entropy-23-01621-t001:** The example frequency of words (selected as conditional attributes).

Attribute Name	True News	Fake News
word_1	608	463
word_2	592	715
word_3	1036	78
word_4	1151	655
word_5	840	585
word_6	631	692
word_7	2193	47
word_8	572	1441
word_9	2520	666
word_10	1227	654
word_11	859	975
word_12	371	970
word_13	471	1167
word_14	577	821
word_15	920	269
word_16	8843	5538
word_17	8369	40
word_18	592	703
word_19	1975	36
word_20	2874	815

**Table 2 entropy-23-01621-t002:** Session attributes.

User ID	Session ID
Day/Month/Year	Hour of begin
Hour of end	Purchase
Total amount	No. products bought
No. references bought	Discount code
New user	Source of navigation
Total time	Time universe (1–7)
No. total pages seen	No. pages universe (1–7) seen
No. universes changes	No. sections changes
No. subsect. changes	No. subsubsect. changes
No. of section seen	No. of subsection seen
No. product pages seen	No. of same product seen

**Table 3 entropy-23-01621-t003:** Discretization procedure for the market indicators. * in the rare cases, where indicator value exceeds the border value (cases with the word “above” or “below”, the indicator value is set to the border value).

Indicator Name	Range *	Discretization Step
Bulls	〈0:0.01〉	0.0005
Bulls	〈−0.01:0〉	0.0005
Bulls	Above 0.1	0.005
Bulls	Below −0.1	0.005
CCI	〈−200:200〉	20.0
DM	〈0:1〉	0.1
OSMA	〈0:0.01〉	0.0005
OSMA	〈−0.01:0〉	0.0005
OSMA	Above 0.1	0.005
OSMA	Below −0.1	0.005
RSI	〈0:100〉	10.0
Stoch	〈0:100〉	10.0

**Table 4 entropy-23-01621-t004:** Information attribute values for fake news data.

Attribute Name	Value Count	Information Attribute
word_1	2	0.998331
word_2	2	0.998048
word_3	2	0.983818
word_4	2	0.996864
word_5	2	0.998051
word_6	2	0.998287
word_7	2	0.957435
word_8	2	0.990542
word_9	2	0.981507
word_10	2	0.996318
word_11	2	0.998106
word_12	2	0.992925
word_13	2	0.992260
word_14	2	0.997342
word_15	2	0.993210
word_16	2	0.986877
word_17	2	0.803497
word_18	2	0.998101
word_19	2	0.961002
word_20	2	0.998470
**Attribute Name**	**Value Count**	**Entropy**
Decision	2	0.998473

**Table 5 entropy-23-01621-t005:** Classification results for fake news data by decision class for full set of attributes [in %] (all bold numbers correspond the best values obtained).

	Decision Tree		Random Forest		Bagging		AdaBoost	
Decision Class	PPV	TPR	PPV	TPR	PPV	TPR	PPV	TPR
FAKE	**98.88**	46.05	93.95	54.87	91.38	**58.67**	92.10	56.88
TRUE	62.70	**99.43**	66.02	96.12	**67.46**	93.93	**66.69**	94.65

**Table 6 entropy-23-01621-t006:** Classification results for fake news data by decision class for limited set of attributes (5 attributes selected) [in %] (all bold numbers correspond the best values obtained).

	Decision Tree		Random Forest		Bagging		AdaBoost	
Decision Class	PPV	TPR	PPV	TPR	PPV	TPR	PPV	TPR
FAKE	**98.88**	46.05	93.68	54.04	93.67	**54.28**	93.67	**54.28**
TRUE	62.70	**99.43**	65.58	96.00	**65.69**	95.97	**65.69**	95.97

**Table 7 entropy-23-01621-t007:** Accuracy results for the classification over fake news data [in %].

	Decision Tree	Random Forest	Bagging	AdaBoost
Accuracy (20 attributes)	71.51	74.55	75.49	74.89
Accuracy (5 attributes)	71.51	74.56	74.17	74.17

**Table 8 entropy-23-01621-t008:** Information attribute values for user websites navigation data.

Attribute Name	Value Count	Information Attribute
No.ofsession	14	0.478533
No.totalpagesseen	65	0.465879
Hourofbegin	101	0.478267
Hourofend	101	0.479259
Day/Month/Year	30	0.474476
Totaltime	77	0.402749
Discountcode	2	0.173014
Newcustomer	2	0.161386
Sourceofnavigation	3	0.479204
No.universeschanges	87	0.364767
TimeUniversequickorder	64	0.347537
TimeUniversestore	74	0.445645
TimeUniverseshoppingcart	61	0.214535
TimeUniversesales	27	0.481156
TimeUniverseconsulting	36	0.481021
TimeUniversecondition	48	0.481082
TimeUniversevarious	74	0.323663
No.pagesuniversequickorder	38	0.339926
No.pagesuniversestore	65	0.432955
No.pagesuniverseshoppingcart	60	0.217707
No.pagesuniversesales	20	0.481144
No.pagesuniverseconsulting	7	0.481129
No.pagesuniversecondition	7	0.481159
No.pagesuniversevarious	64	0.349787
No.subsectionsseen	16	0.439947
No.ofsectionchanges	47	0.464543
No.productpagesseen	64	0.478707
No.ofsameproductseen	42	0.468668
No.ofsubsectionseen	88	0.436594
No.subsectionchanges	78	0.471988
No.subsubsectionchanges	89	0.470698
**Attribute Name**	**Value Count**	**Entropy**
Purchase(Decision)	2	0.481233

**Table 9 entropy-23-01621-t009:** Classification results for user websites navigation data by decision class values for full set of attributes [in %].

	Decision Tree		Random Forest		Bagging		AdaBoost	
Decision Class	PPV	TPR	PPV	TPR	PPV	TPR	PPV	TPR
Purchase=yes	99.91	89.50	99.92	87.44	49.46	92.55	89.15	91.49
Purchase=no	98.80	99.99	98.56	99.99	99.03	89.03	99.01	98.70

**Table 10 entropy-23-01621-t010:** Classification results for user websites navigation data by decision class values for limited set of attributes (7 attributes selected) [in %].

	Decision Tree		Random Forest		Bagging		AdaBoost	
Decision Class	PPV	TPR	PPV	TPR	PPV	TPR	PPV	TPR
Purchase=yes	99.92	89.50	99.92	89.50	96.70	92.42	98.83	90.61
Purchase=no	98.80	99.99	98.80	99.99	99.13	99.63	98.92	99.88

**Table 11 entropy-23-01621-t011:** Accuracy results for user websites navigation data [in %].

	Decision Tree	Random Forest	Bagging	AdaBoost
Accuracy (31 attributes)	98.90	98.69	89.40	97.96
Accuracy (7 attributes)	98.90	98.90	98.89	98.91

**Table 12 entropy-23-01621-t012:** Information attribute values for real estate market data.

Attribute Name	Value Count	Information Attribute
Builddate	166	3.294200
Totalnumberoffloors	35	3.418894
Buildingmaterial	6	3.529588
Floornumber	14	3.536040
Area	3,698	2.071138
Buildingtype	5	3.455887
Conditionstate	6	3.485657
Windows	3	3.542212
Privatead	2	3.576844
Markettype	2	3.552212
Ownershiptype	2	3.572712
**Attribute Name**	**Value Count**	**Entropy**
Pricebucket(decision)	16	3.579787

**Table 13 entropy-23-01621-t013:** Classification results for real estate market data by decision class for full set of attributes [in %] (all bold numbers correspond the best values obtained).

	Decision Tree		Random Forest		Bagging		AdaBoost	
Decision Class	PPV	TPR	PPV	TPR	PPV	TPR	PPV	TPR
1	**40.00**	9.52	0.00	0.00	25.00	**23.80**	0.16	**23.81**
2	0.00	0.00	**100.0**	0.00	99.47	**99.68**	24.61	10.02
3	0.00	0.00	0.00	0.00	**99.27**	**99.85**	0.00	0.00
4	0.00	0.00	0.00	0.00	**99.12**	**99.47**	0.00	0.00
5	0.00	0.00	0.00	0.00	**99.23**	**100.0**	0.00	0.00
6	0.00	0.00	0.00	0.00	**97.24**	**99.64**	0.00	0.00
7	0.00	0.00	49.08	21.62	**94.47**	**94.47**	27.44	87.80
8	0.00	0.00	0.00	0.00	**79.01**	**60.95**	0.67	9.52
9	0.00	0.00	0.00	0.00	**93.09**	**92.33**	0.00	0.00
10	99.57	**100.00**	**100.00**	7.10	99.57	99.74	99.57	79.98
11	0.00	0.00	0.00	0.00	**98.97**	**100.0**	0.00	0.00
12	21.55	**100.00**	31.25	21.79	**99.91**	**100.0**	67.78	10.03
13	**99.85**	**100.00**	32.13	91.97	**99.85**	99.90	0.00	0.00
14	**99.82**	0.99	73.49	**100.0**	**99.82**	99.65	38.48	**100.0**
15	**100.00**	**100.00**	87.70	**100.0**	**100.0**	**100.0**	39.41	9.97

**Table 14 entropy-23-01621-t014:** Classification results for real estate market data by decision class for a limited set of attributes (3 attributes selected) [in %]. (all bold numbers correspond the best values obtained).

	Decision Tree		Random Forest		Bagging		AdaBoost	
Decision Class	PPV	TPR	PPV	TPR	PPV	TPR	PPV	TPR
1	0.00	0.00	0.00	0.00	**27.78**	**23.81**	0.16	**23.81**
2	0.00	0.00	56.27	15.61	**99.58**	**99.79**	24.61	10.02
3	0.00	0.00	0.00	0.00	**99.14**	**100.00**	0.00	0.00
4	0.00	0.00	0.00	0.00	**98.79**	**99.82**	0.00	0.00
5	0.00	0.00	0.00	0.00	**99.24**	**100.00**	0.00	0.00
6	0.00	0.00	0.00	0.00	**96.91**	**99.64**	0.00	0.00
7	0.00	0.00	0.00	0.00	**89.01**	**90.89**	27.44	87.80
8	0.00	0.00	0.00	0.00	**60.38**	**30.48**	0.67	9.52
9	0.00	0.00	0.00	0.00	**89.37**	**91.53**	0.00	0.00
10	**99.57**	100.00	98.91	15.57	**99.57**	**99.66**	**99.57**	79.98
11	0.00	0.00	0.00	0.00	**98.79**	**99.82**	0.00	0.00
12	21.56	**100.00**	39.43	30.61	**99.92**	99.92	67.78	10.03
13	**99.85**	**100.00**	30.61	99.95	**99.85**	99.74	0.00	0.00
14	99.74	**100.00**	94.89	**100.00**	99.74	**99.82**	38.49	**100.00**
15	**100.00**	**100.00**	98.98	**100.00**	**100.00**	**100.00**	39.55	10.03

**Table 15 entropy-23-01621-t015:** Accuracy results for the classification over real estate data [in %].

	Decision Tree	Random Forest	Bagging	AdaBoost
Accuracy (15 attributes)	69.00	53.69	99.07	28.92
Accuracy (3 attributes)	69.00	56.71	98.71	28.92

**Table 16 entropy-23-01621-t016:** Information attribute values for sport data.

Attribute Name	Value Count	Information Attribute
	Germany	
Season	10	1.535055
Round	30	1.514509
Team1(Team2)	28 (28)	1.471725 (1.464826)
PositionT1(T2)	18 (18)	1.417129 (1.438857)
MatchesT1(T2)	30 (30)	1.514509 (1.514509)
WinningsT1(T2)	30 (30)	1.461058 (1.469589)
DrawsT1(T2)	16 (16)	1.506093 (1.502721)
LosersT1(T2)	24 (25)	1.473808 (1.468121)
GoalsscoredT1(T2)	92 (94)	1.457664 (1.454900)
GoalsconcededT1(T2)	74 (77)	1.476410 (1.470156)
GoaldifferenceT1(T2)	116 (117)	1.382856 (1.395159)
PointsT1(T2)	85 (86)	1.440930 (1.452137)
SeriesT1(T2)	40 (40)	1.505347 (1.498025)
Match Result (Decision)	3	1.539089
	Italy	
Season	10	1.538402
Round	34	1.531829
Team1(Team2)	34 (34)	1.458934 (1.460329)
PositionT1(T2)	20 (20)	1.415896 (1.424320)
MatchesT1(T2)	34 (34)	1.531829 (1.531829)
WinningsT1(T2)	33 (32)	1.468639 (1.475596)
DrawsT1(T2)	19 (19)	1.514713 (1.512888)
LosersT1(T2)	29 (30)	1.469099 (1.465326)
GoalsscoredT1(T2)	92 (91)	1.481346 (1.475606)
GoalsconcededT1(T2)	86 (86)	1.484403 (1.488688)
GoaldifferenceT1(T2)	112 (110)	1.397699 (1.398332)
PointsT1(T2)	97 (97)	1.448064 (1.454139)
SeriesT1(T2)	40 (40)	1.506473 (1.516376)
**Attribute Name**	**Value Count**	**Entropy**
MatchResult(Decision)	3	1.545029
	Spain	
Season	10	1.519782
Round	34	1.514018
Team1(Team2)	33 (33)	1.438616 (1.436341)
PositionT1(T2)	20 (20)	1.403615 (1.401159)
MatchesT1(T2)	34 (34)	1.514018 (1.514018)
WinningsT1(T2)	32 (32)	1.451360 (1.455801)
DrawsT1(T2)	19 (19)	1.493165 (1.486429)
LosersT1(T2)	27 (27)	1.462337 (1.453162)
GoalsscoredT1(T2)	115 (111)	1.448292 (1.437169)
GoalsconcededT1(T2)	82 (81)	1.467792 (1.464831)
GoaldifferenceT1(T2)	130 (132)	1.376633 (1.375712)
PointsT1(T2)	95 (94)	1.438554 (1.435251)
SeriesT1(T2)	40 (40)	1.493445 (1.488321)
**Attribute Name**	**Value Count**	**Entropy**
MatchResult(Decision)	3	1.523545

**Table 17 entropy-23-01621-t017:** Classification results for sport data by decision class for full set of attributes [in %] (all bold numbers correspond the best values obtained).

	Decision Tree		Random Forest		Bagging		AdaBoost	
Decision Class	PPV	TPR	PPV	TPR	PPV	TPR	PPV	TPR
			Germany					
1	53.66	85.51	53.96	**88.01**	56.94	70.36	**58.71**	68.44
2	55.74	48.34	**58.74**	**51.66**	51.92	49.94	55.05	50.92
*X*	**38.18**	3.18	0.00	0.00	33.09	20.45	**38.31**	**30.30**
			Italy					
1	54.45	87.44	54.57	**89.79**	**60.39**	70.85	58.90	69.58
2	56.56	52.63	**59.12**	53.13	52.04	55.81	50.17	**57.10**
*X*	45.16	1.62	**100.00**	0.46	34.40	**21.21**	39.87	**20.97**
			Spain					
1	55.78	88.11	55.57	**90.12**	**59.72**	69.23	**60.34**	69.79
2	52.00	**46.88**	**54.85**	44.87	49.55	**46.98**	50.73	40.53
*X*	0.00	0.00	**58.33**	0.85	30.29	22.83	31.73	**29.47**

**Table 18 entropy-23-01621-t018:** Accuracy results for the classification over sport data [in %].

	Decision Tree	Random Forest	Bagging	AdaBoost
Germany	53.89	55.24	51.83	53.70
Accuracy 24 attributes				
Germany	53.31	54.84	49.01	55.78
Accuracy 6 attributes				
Italy	54.96	55.85	53.59	53.35
Accuracy 24 attributes				
Italy	54.47	54.85	50.09	53.54
Accuracy 6 attributes				
Spain	54.82	55.41	51.55	51.64
Accuracy 24 attributes				
Spain	55.01	55.49	51.74	55.31
Accuracy 6 attributes				

**Table 19 entropy-23-01621-t019:** Classification results for sport data by decision class values for for limited set of attributes (6 attributes selected) [in %] (all bold numbers correspond the best values obtained).

	Decision Tree		Random Forest		Bagging		AdaBoost	
Decision Class	PPV	TPR	PPV	TPR	PPV	TPR	PPV	TPR
			Germany					
1	53.43	84.93	54.28	**86.59**	56.06	65.86	**57.08**	80.27
2	54.53	48.83	**56.27**	52.40	49.08	49.20	54.77	**60.76**
*X*	22.22	1.21	0.00	0.00	26.62	**18.06**	**40.24**	5.01
			Italy					
1	54.47	86.83	53.73	**89.05**	**58.12**	67.61	55.92	80.31
2	55.76	54.32	**57.96**	51.34	49.85	50.94	52.44	**55.51**
*X*	2.00	0.12	0.00	0.00	27.26	**18.89**	**28.39**	5.10
			Spain					
1	55.98	87.78	55.88	**89.48**	**60.12**	71.28	58.03	85.14
2	52.24	48.15	**54.28**	46.98	49.23	47.62	51.08	**50.26**
*X*	0.00	0.00	0.00	0.00	27.84	**18.96**	**31.68**	3.86

**Table 20 entropy-23-01621-t020:** Information attribute values for the financial data (all bold numbers correspond the best values obtained).

Attribute Name	AUDUSD		EURUSD		GBPUSD		NZDUSD	
	Value	Inf.	Value	Inf.	Value	Inf.	Value	Inf.
	Count	Attribute	Count	Attribute	Count	Attribute	Count	Attribute
SMA14	2714	**0.077183**	2717	**0.078407**	2749	**0.050490**	2665	**0.098638**
SMA50	2686	**0.087042**	2673	**0.094808**	2734	**0.057822**	2653	**0.119658**
$SMA14^{\prime }$	689	1.380616	767	1.258658	853	1.103368	623	1.417858
$SMA50^{\prime }$	440	1.612055	497	1.467387	510	1.355281	389	1.652498
Bulls	341	1.747386	336	1.654923	296	1.584065	290	1.800077
CCI	78	1.936437	76	1.852890	80	1.752818	80	1.964629
DM	10	2.008175	11	1.928776	10	1.807938	9	2.035717
OSMA	154	1.899572	166	1.812773	214	1.653933	131	1.939371
RSI	5	2.014515	5	1.931348	6	1.812072	5	2.039597
Stoch	10	2.008801	11	1.928170	12	1.807163	11	2.034959
**Attribute Name**	**Value** **Count**	**Entropy**	**Value** **Count**	**Entropy**	**Value** **Count**	**Entropy**	**Value** **Count**	**Entropy**
Decision	5	2.321928	5	1.935982	5	1.815961	5	2.044563

**Table 21 entropy-23-01621-t021:** Classification results for the financial data by decision class for full set of attributes [in %].

	Decision Tree		Random Forest		Bagging		AdaBoost	
Decision Class	PPV	TPR	PPV	TPR	PPV	TPR	PPV	TPR
AUDUSD								
BUY	4.05	1.53	-	-	1.52	2.55	-	-
SELL	-	-	-	-	1.99	3.23	6.25	0.46
STRONGBUY	35.55	34.77	33.22	49.91	31.03	30.81	33.15	37.32
STRONGSELL	35.08	53.50	30.57	32.15	26.75	21.89	35.76	49.59
WAIT	-	-	-	-	1.29	0.93	5.88	0.47
EURUSD								
BUY	-	-	-	-	1.83	2.30	2.22	0.57
SELL	-	-	-	-	1.44	1.62	-	-
STRONGBUY	33.01	20.79	29.45	19.65	35.46	38.43	34.06	35.28
STRONGSELL	38.06	66.58	37.71	65.08	36.43	34.42	35.80	44.44
WAIT	-	-	-	-	1.49	0.61	4.76	0.61
GBPUSD								
BUY	-	-	-	-	2.27	0.88	-	-
SELL	-	-	-	-	1.56	2.05	-	-
STRONGBUY	42.92	84.02	46.09	65.25	42.82	48.61	43.09	51.39
STRONGSELL	44.44	16.84	47.45	43.72	41.20	40.00	44.17	49.72
WAIT	-	-	-	-	2.22	0.67	-	-
NZDUSD								
BUY	-	-	-	-	5.95	4.81	12.82	2.40
SELL	-	-	-	-	5.21	4.80	7.27	1.75
STRONGBUY	40.20	85.76	39.35	83.06	37.41	45.94	40.11	62.01
STRONGSELL	39.63	13.94	39.29	16.25	33.80	31.02	38.48	35.00
WAIT	-	-	-	-	5.81	2.50	-	-

**Table 22 entropy-23-01621-t022:** Classification results for the financial data by the decision class values for 25% of attributes with the lowest information attribute (in %).

	Decision Tree		Random Forest		Bagging		AdaBoost	
Decision Class	PPV	TPR	PPV	TPR	PPV	TPR	PPV	TPR
AUDUSD								
BUY	2.74	1.02	4.44	1.02	2.33	3.06	2.33	0.51
SELL	-	-	-	-	2.95	4.61	-	-
STRONGBUY	33.43	29.69	32.32	36.56	33.85	34.45	36.80	44.23
STRONGSELL	34.89	56.49	31.99	44.60	29.72	23.89	32.94	43.60
WAIT	-	-	-	-	4.85	5.12	-	-
EURUSD								
BUY	-	-	-	-	2.97	3.45	-	-
SELL	-	-	-	-	3.91	5.41	-	-
STRONGBUY	32.11	21.50	34.14	28.23	38.10	38.20	30.41	24.21
STRONGSELL	39.37	67.34	37.81	59.23	41.24	38.76	36.59	58.48
WAIT	-	-	-	-	3.73	3.05	-	-
GBPUSD								
BUY	-	-	-	-	2.70	2.65	16.67	0.88
SELL	-	-	-	-	2.11	3.42	-	-
STRONGBUY	41.28	37.05	44.40	54.92	42.92	42.95	42.87	48.77
STRONGSELL	42.52	60.73	43.32	47.53	42.21	41.05	44.39	52.55
WAIT	-	-	-	-	-	-	-	-
NZDUSD								
BUY	3.51	0.96	-	-	1.27	1.44	-	-
SELL	-	-	-	-	6.46	8.30	27.27	1.31
STRONGBUY	39.61	84.97	38.87	82.60	37.45	38.72	40.69	70.28
STRONGSELL	39.13	11.63	42.26	16.90	29.32	27.89	39.56	30.10
WAIT	-	-	-	-	2.50	1.50	5.13	1.00

**Table 23 entropy-23-01621-t023:** Accuracy results for the classification over the financial data [in %].

	Decision Tree	Random Forest	Bagging	AdaBoost
AUDUSD	34.45	32.15	21.12	33.93
AUDUSD 2 atr.	33.55	31.70	23.78	34.32
EURUSD	36.13	35.04	30.02	32.74
EURUSD 2 atr.	36.73	36.03	32.19	34.11
GBPUSD	43.04	46.63	38.12	43.32
GBPUSD 2 atr.	41.97	43.89	36.28	43.47
NZDUSD	39.55	39.34	30.99	38.32
NZDUSD 2 atr.	38.41	39.39	26.89	39.63

**Table 24 entropy-23-01621-t024:** Number of attributes after selection.

	CFS	Proposed Approach	Original
Fake News Data	8	5	20
User Websites Navigation Data	5	7	31
Real-Estate Market Data	9	3	15
Sport Data (Germany)	8	6	24
Financial Data (GBPUSD)	2	2	11

**Table 25 entropy-23-01621-t025:** Accuracy results for the classification over the data after selection [in %].

Data	Decision Tree	Random Forest	Bagging	AdaBoost
FN	71.51	74.24	74.25	74.27
Change:	—	−0.32	+0.08	+0.10
UWN	98.90	98.90	98.60	98.81
Change:	—	—	−0.29	−0.10
R-EM	69.00	57.67	98.93	28.92
Change:	—	+0.96	+0.22	—
SD	53.87	55.67	50.39	55.89
Change:	+0.56	+0.83	+1.38	+0.10
FD	42.44	41.55	33.52	43.51
Change:	+0.47	−2.34	−2.76	+0.04

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
