# Peer review of "Real-World Data Difficulty Estimation with the Use of Entropy"

_entropy, 2021, doi:10.3390/e23121621_

Round 1

Reviewer 1 Report

In the article, the authors present the techniques of preparing real data sets for further analysis, which is the classification in the framework of supervised machine learning area. The conducted research also concerns the problems related to the selection of features and the ranking of attributes based on the measure of entropy.

The article needs some improvement.

  1. In the related work section, the authors have described many applications of the entropy measure, however, it would be helpful to briefly describe what entropy is and why the authors chose this measure.
  2. In some places, the article lacks a more precise description of concepts, e.g. data is presented in the form of a decision table, but in equation (2) the meaning of the word symbol has not been related to the data presented in the form of a decision table.

n - based on line 136 is the number of attributes, I suppose |DS| is the number of objects - however, it has not been specified previously.

pi is the proportion of i-th symbol relative to what?

  1. In the experimental part, the authors present the classification results for the entire set of attributes and for a selected subset of attributes. Therefore, you should consider supplementing the article with a short section on the selection of features and ranking of attributes.
  2. Presentation and description of a general scheme for the preparation of data that has been subjected to classification and analysis would allow for the systematization of the preprocessing steps. Maybe before subsection 4.1 description of main steps of preprocessing conducted for all datasets would be useful.
  3. During the selection of the features, the authors selected 25% of the attributes from the entire set, it is interesting what is the minimum number of attributes ensuring the same level of classification as the entire set of features.
  4. The authors can indicate future directions of research

Typos:

log(pi) – lower index in (2)

Table 10 – for for

Author Response

All responses for remarks are attached as a file. 

Reviewer 2 Report

Dear Authors,

The submitted paper “Real-world Data Difficulty Estimation with the Use of Entropy” is addressing an important and interesting topic, therefore thank you very much for your work and the contribution.

The paper discuss the problem of deriving nonredundant information. The Authors investigate the possibilities of using the entropy measure as an indicator of data difficulty. The Auhors contribution is two folds: inconsistent data preprocessing and using the entropy-based measure to capture the nonredundant noncorrelated core information from the data. From formal point of view the Authors used well known algorithms and methods. Even though the Authors do not introduce a new method and work on the basis of known approaches from a methodical point of view the paper scores high. Results are very interesting and give strong contribution in the subject under investigation. Paper is well written. Literature is relevant. Generally, paper is well structured, important theoretical and  practical aspects of the examined problem are studied and presented in a clear and consistent manner. Practically, paper is well positioned in Entropy journal’s aim and scope.

To sum up - paper interesting and technically sound. Thus I suggest to accept the paper.

Author Response

We want to thank an anonymous reviewer for the comments on the article. There were no changes in the paper related directly to this review; however, please note that some text fragments were changed according to suggestions made by two remaining reviewers. These changes are marked with blue (related to Review 1) and green (related to Review 3).

Reviewer 3 Report

Research on data difficulty is an important research direction.  The authors calculated entropy of different datasets, and try to find out the possibility of using entropy to measure the data difficulty of datasets,or so called, I quote "explore the possibility of using entropy as an indicator of data difficulty" 

Using entropy as a measurement is not quite new idea, authors should search through academic articles on data difficulty and feature selecting, And do a better literature review in the introduction part.

At the same time, data difficult is a general term, decomposition of the minority class concept into many sub-concepts, class overlapping are all can be called data difficulty, a detailed research on how these specific factors related to entropy will be a good choice. 

About the writing,

1, Line 9, "However, such actions...." This transition statement doesn't read very smoothly in context.

2. Line 54,55, using entropy to estimate entropy based..., seems wired and difficult to be understood.

3.  the Methodology part , the symbols system is a little be messy, though I can understand what the authors want to describe, but the symbols are defined fairly loose. The authors should rethought about it and rewrite it.

4. It takes too much words to describe the datasets without the characteristics analysis on these data.

Author Response

(The authors gave the same response as above.)

Round 2

Reviewer 3 Report

No more conmments